# Dynamic Modal Analyses of Building Structures Employing Site-Specific Response Spectra Versus Code Response Spectrum Models

**Prashidha Khatiwada** [1,*] **, Yiwei Hu** [1] **, Elisa Lumantarna** [1] **and Scott J. Menegon** [2]

1   Department of Infrastructure Engineering, The University of Melbourne, Melbourne, VIC 3010, Australia
2   Centre for Sustainable Infrastructure and Digital Construction, Swinburne University of Technology, Melbourne, VIC 3122, Australia
*   Correspondence: khatiwadap@unimelb.edu.au; Tel.: +61-3-9035-5511

**Abstract:** This paper is aimed at giving structural designers guidance on how to make use of elastic site-specific response spectra for the dynamic modal analysis of a structure in support of its structural design. The use of response spectra in support of the pushover analysis of an RC building forming part of the non-linear static analysis procedure (that can be used to predict seismic demand without relying on the code-stipulated default R factor) is also presented. Seismic analysis of structures based on the use of site-specific response spectra can help to achieve a more optimised, and cost-effective, structural design compared to the conventional approach employing a response spectrum model stipulated by the code for different site classes. Currently, the methodology is only adopted in major projects in which enough resources are available to engage experts who are skilled in operating the procedure; thus, the use of site-specific response spectra in structural engineering practice is still limited despite the merits of the procedure. Deriving a site-specific response spectrum requires a database of representative ground motion records to be developed. Extra analytical tasks to be undertaken include the processing of bore log data, site response analyses, and selection/scaling of bedrock accelerograms for input into site response analyses. Guidelines for implementing this design methodology are currently lacking. To promote the wide adoption of site-specific seismic design, this article presents the procedure for developing the required site-specific design spectra, as well as guidelines for applying these spectra for seismic design based on analyses of linear, or nonlinear, models of the building. Non-linear analysis can be accomplished by dealing with macroscopic models as illustrated in a case study.

**Keywords:** site-specific structural analysis; nonlinear response spectrum analysis; nonlinear analysis; reinforced concrete buildings

## 1. Introduction

With the conventional code stipulated force-based procedure, the design seismic actions are mainly determined by predicting the elastic strength demand on the structure along with a modification factor (which is also known as the behaviour factor or strength reduction factor) to account for the post-elastic behaviour of the structure when subject to severe ground shaking. Either equivalent static analyses, or dynamic analyses, involving the use of a design response spectrum (or time-history analysis involving the use of accelerograms), are employed for determining the elastic strength demands on the structure. The equivalent static analysis method is rarely used for the following reasons: (1) limitations of the scope of application of the method to low-rise buildings that are completely free of irregularities, which is uncommon; (2) inherent over-conservatism with the estimates of the base shear of the building resulted from taking the total mass of the building as the effective mass; and, (3) neglecting effects of the higher modes of vibrations. Of the two types of dynamic analysis methods, the response spectrum analysis method as opposed to the

time-history analysis method is preferred by design engineers, as it is more straightforward to implement and is computationally inexpensive. With response spectrum analyses, only the peak response quantity as read off, or inferred, from the design response spectrum is of interest to the structural designer. This article deals with the response spectrum analysis method.

With the response spectrum analysis method, the structural design engineer is required to use either a code response spectrum stipulated for different soil classes (or ground types) or a site-specific response spectrum, which has been derived from the site response analysis of a representative soil column model. A comparison of the two types of response spectra presented in the acceleration (RSA) format based on compliance with the Australian Standard for Seismic Actions [1] is shown in Figure 1.

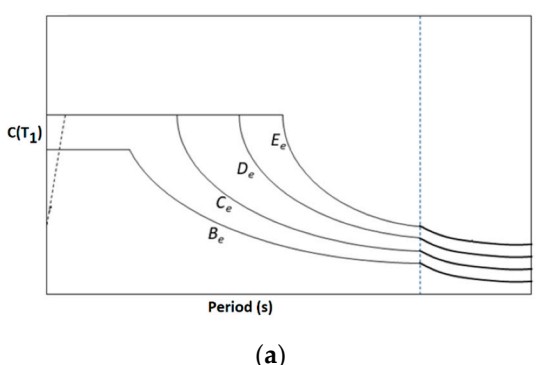

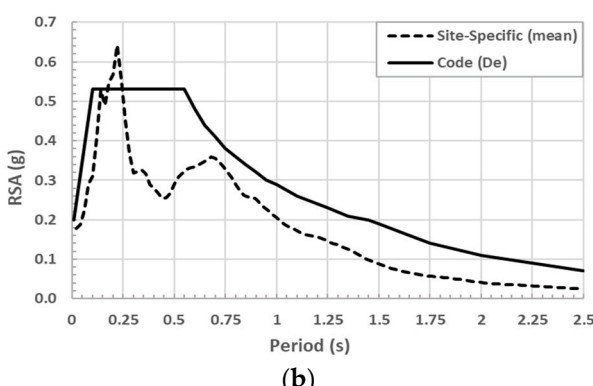

(**a**)                                                            (**b**)

**Figure 1.** Comparison of the site-specific mean response spectrum and the code response spectrum. (**a**) Elastic code response spectrum for different ground types. (**b**) Site-specific (mean) and the code acceleration response spectrum.

The use of a site-specific response spectrum is found to have considerable benefits over the code response spectra because of the better accuracy over the code model in accounting for the soil-structure resonance phenomena [2]. Despite the above-mentioned benefits, site-specific response spectra are rarely used in design practice because of the need to generate site-specific response spectra, which can be a very cumbersome process and is filled with uncertainties [3–5]. Recent publications by the authors [5–7] have gone a long way in having these uncertainties resolved by providing detailed guidance. In this article, the determination of the design site-specific response spectrum–through averaging multiple response spectra as obtained from analyses of the generated soil surface accelerograms—is illustrated in detail with a case study (refer to Section 2).

The response spectrum method of seismic analysis as described above is found on force-based principles (refer to Section 3 for details). There are known shortcomings with this conventional approach in the prediction of the seismic performance of the structure and modelling damage incurred in a projected earthquake scenario. The shortcomings are much to do with the use of a generalised reduction factor to allow for ductility in the structure and the adoption of an initial/elastic stiffness of the structural elements. The later part of the article (Section 4) presents recommendations for replacing the conventional procedure with a nonlinear analysis procedure, which involves the use of a rational method for accurately modelling ductile behaviour and deformation (stiffness-related) behaviour of the structure [8–11]. Two nonlinear analysis methods: (1) pushover analysis and (2) time-history analysis can be employed for this purpose. Pushover analysis, which is initially recommended by FEMA [9], employs a straightforward and computationally inexpensive procedure for predicting the nonlinear response behaviour of a low to medium-rise building [12]. Existing pushover procedures can be classified into two types: Detailed procedures and simplified procedures. The detailed pushover analysis procedure requires the use of specific computer software to predict the inelastic response behaviour of the structure incorporating the provisions of plastic hinges. The simplified pushover analysis

procedure, which is widely recognised, involves modelling the building taking into account the formation of the plastic hinges at the base of the lateral resisting elements. Different studies in the literature have suggested simplified models for predicting the pushover behaviour of rectangular shear walls in low to medium-rise RC buildings with reasonable accuracy [13–15]. The use of a simplified pushover analysis procedure, based on the original model given by Menegon [15], is illustrated in Section 4 to demonstrate its application with a case study where a site-specific response spectrum is used to represent the seismic actions. The presentation of time-history analyses involving the use of accelerograms is beyond the scope of this paper.

## 2. Generation of Site-Specific Response Spectra

Major codes of practice including the Australian Standard for Seismic Actions allow the use of site-specific response spectra in seismic design to supplant code design response spectra. Site-specific spectra need to be generated from bedrock ground motions that have been subject to modifications along the path of seismic wave transmission to the ground surface. The nonlinear or equivalent linear analysis of the soil column model is to take into account the properties of each soil layer as determined from laboratory testing, or the use of the relevant published models. The rock outcrop motions are to be sourced from the NGA-West2 strong motion database, which can be accessed through the website of the Pacific Earthquake Engineering Research (PEER) Center [16]. Pre-defined ranges of the earthquake magnitude and site-fault distances are to be specified when extracting accelerograms from the database. The prescribed earthquake scenarios should be consistent with a pre-defined design return period in an intraplate environment.

In recent studies undertaken by the authors [5–7], a simplified procedure for generating code-compliant site-specific response spectra has been developed. The developed procedure, which is presented in the schematic diagram of Figure 2, is easy to implement. An online tool for delivering the response spectrum generation as described has been mounted on the website "quakeadvice.org" [17].

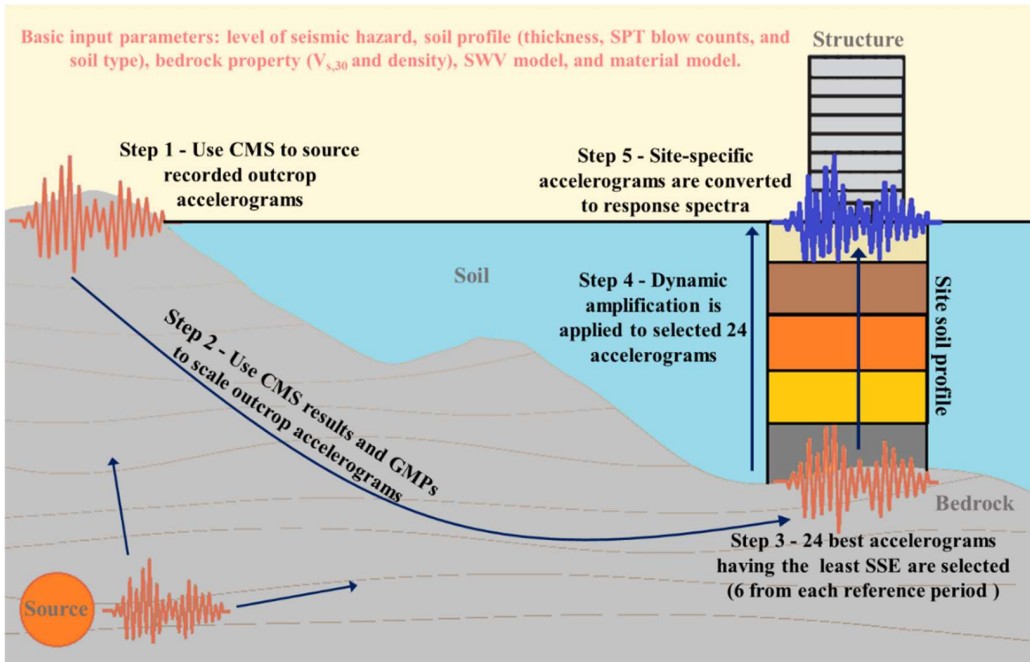

**Figure 2.** Procedures for generating site-specific response spectra. The orange and blue colour represents accelerograms at the bedrock (material in grey colour), and soil surface, respectively. The soil column profile obtained from the borehole is represented by layers of five different colours.

The response spectrum generation procedure can be summarised in five steps as outlined below:

Step 1: The conditional mean spectrum (CMS) approach is adopted for sourcing accelerograms [18–20]. Separate CMS for matching the code spectrum on rock sites at four reference periods (T*), e.g., 0.2 s, 0.5 s, 1 s, and 2 s are to be constructed, which cover a broad range of periods to ensure that an envelope of the mean spectra from each reference period collectively cover (approximately) the entire bedrock code spectrum (i.e., Class Be for AS 1170.4). For each CMS, the controlling magnitude-distance (MR) combinations are first identified by hazard disaggregation analyses for a pre-determined design return period, which defines the intensity of the seismic hazard.

Step 2: For each earthquake scenario (expressed in terms of the M-R combinations) as identified in Step One, calculate the weighted median and standard deviation of response spectral acceleration as predicted from the ground motion prediction equations (GMPEs) that are representative of the region. Values of the standard deviation and correlation coefficients are applied to scale the CMS to match the code spectrum at each of the reference periods.

Step 3: Accelerograms representing rock outcrop motions conforming to the considered M-R combinations, fault type, and rock shear wave velocity ($V_{s,30}$) are to be extracted from the PEER database. The accelerograms need to be scaled so that their calculated response spectra match closely with the CMS in the period range of 0.2T* to 2T*. With each CMS, the "best six" accelerograms having the least sum of squares error (SSE) are to be selected.

Step 4: Twenty-four accelerograms (six from each of the four reference periods) representing motions on the soil surface are obtained by subjecting the soil column model of the site-to-site response analysis (using the scaled rock outcrop accelerograms as input into the analyses). Information to be processed includes the thickness of each soil layer, the standard penetration blow counts (SPTs), and soil types. Imai and Tonouchi [21] SWV model may be employed for converting SPTs to shear wave velocity (SWV) values for each soil layer for input into the site response analysis. The Hardin and Drnevich [22] material model is commonly used in the analysis for modelling the degradation of the soil shear modulus and damping for each soil type.

Step 5: The generated soil surface accelerograms are then subject to time-step integration [23] for determining their respective response spectra. The response spectra associated with each reference period are then averaged to obtain the respective mean response spectra. Thus, there are four mean site-specific response spectra for use in structural analyses of the building. Every mean spectrum must fulfil the requirements of codes of practice, which typically requires averaging across at least five site-specific response spectra.

The application of the procedure is demonstrated herein through a case study building that is located in a stable continental region such as Australia. The design seismic hazard is characterised by a hazard design factor (Z) of 0.08 and probability factor ($k_p$) of 1.8 (corresponding to a 2500-year return period). The soil column model that was derived from the subsoil geotechnical investigation of the site is presented in Table A1. The screenshot of the input parameters into the site response analysis as defined in "quakeadvice.org" is presented in Figure A1. The natural period of the example site is estimated at 0.61 s, which corresponds to the site classification of $D_e$ in accordance with AS 1170.4-2007 [1]. The procedure for calculating the site natural period is outlined in the commentary to AS 1170.4 [24]. The twenty-four accelerogram records that were sourced from the PEER database and then scaled to fit with the CMS using tools built into "quakeadvice.org" [17] are listed in Table 1. Details to be specified when sourcing accelerograms from the database include the style of (reverse/oblique) faulting, magnitude range (half-bin width) of $\pm 0.3\,M_w$ Joyner–Boore distance range (half-bin width) of $\pm 30$ km centred at the distance of the controlling scenarios, and $V_{s,30}$ of the rock outcrop of 450–1800 m/s. Accelerogram record nos.: 1–6, 7–12, 13–18, and 19–24 as listed in Table 1 correspond to reference periods of 0.2 s, 0.5 s, 1 s, and 2 s, respectively. The twenty-four individual and four mean site-

specific response spectra obtained from the five-step procedure as outlined above are presented in the velocity (RSV) format in the unit of mm/s in Figure 3. Note, the value of RSV was calculated based on applying the conversion: RSV = RSA × (T/2π). The four mean response spectra for each reference period for the bedrock ground motions and the site-specific ground motions in the acceleration (RSA) format are shown in Figure 4a,b, respectively. The respective $B_e$ and $D_e$ site spectra as per AS 1170.4-2007 [1] are also shown in Figure 4a,b for comparison. It can be seen in Figure 4a that the envelope of the four reference period mean spectra for the bedrock ground motions approximately cover the entire code spectrum (as per the Step 1 requirement). Further, it can be seen in Figure 4b that there is a low point (relative to the code spectrum) in the mean site-specific spectra around 0.5 s, since the site-specific spectra are based on the actual site period (which is 0.61 s in the initial state and shifted to 0.7–0.82 s under the applied motions) whereas the code spectrum is meant to cover all site periods. These design response spectra are to be used in response spectrum analysis of the case study buildings as presented in Sections 3 and 4.

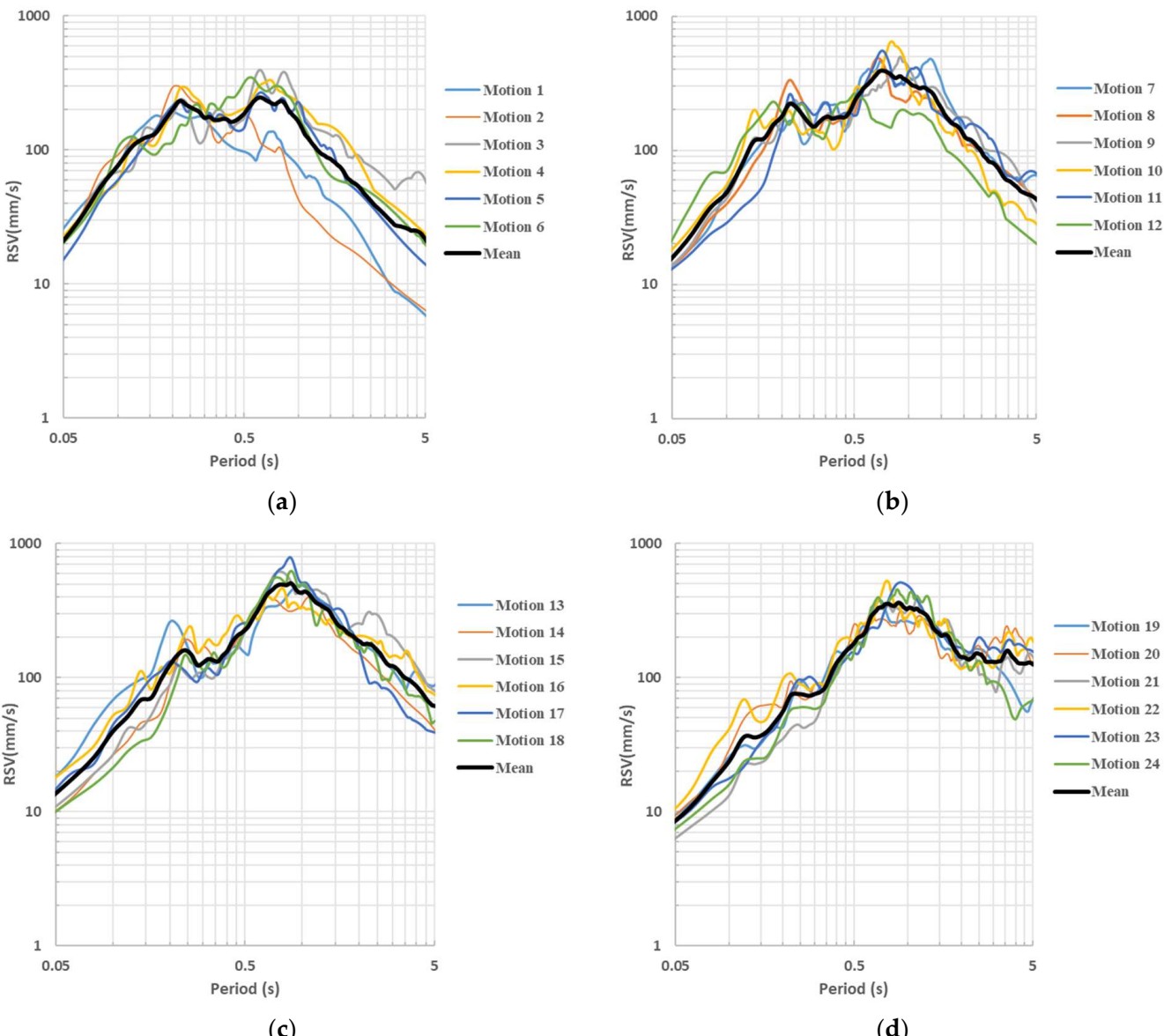

**Figure 3.** Individual and mean site-specific velocity response spectra for the four reference periods: (**a**) site-specific 1 (T* = 0.2 s), (**b**) site-specific 2 (T* = 0.5 s), (**c**) site-specific 3 (T* = 1 s), and (**d**) site-specific 4 (T* = 2 s).

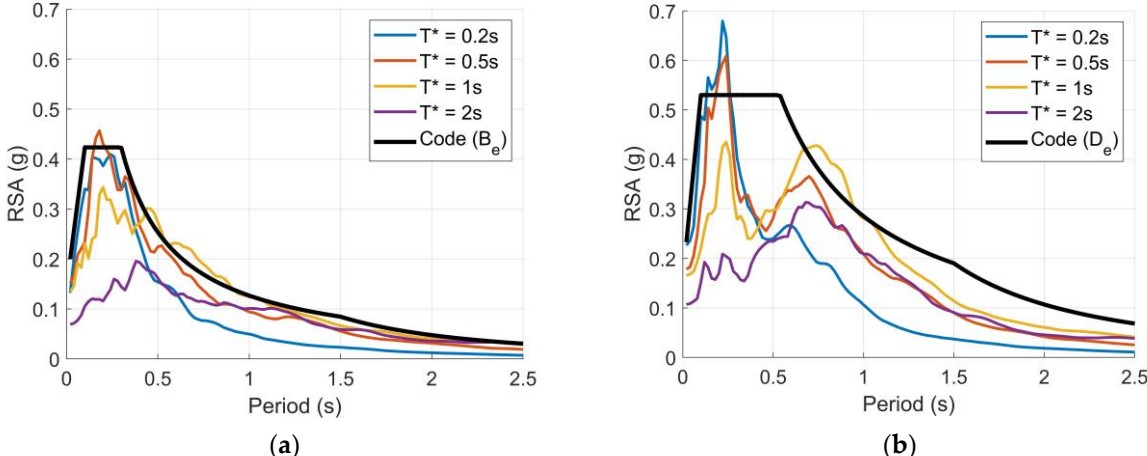

**Figure 4.** Acceleration response spectrum (RSA) as stipulated in AS 1170.4-2007 [1] for: (**a**) $B_e$ site compared to bedrock response spectra (based on the Table 1 ground motions), and (**b**) $D_e$ site compared to site-specific mean response spectra generated from this study.

**Table 1.** Detail of the 24 earthquake records selected from the PEER database.

| Ref. Period | Earthquake Name | Reference Periods (s) | Year | Station Name | $M_w$ | $R_{jb}$ (km) | PGA (g) | Scaling Factor |
|---|---|---|---|---|---|---|---|---|
| 1 | Whittier Narrows-02 | 0.2 | 1987 | Mt Wilson—CIT Seis Sta | 5.27 | 16.5 | 0.175 | 1.21 |
| 2 | Northridge-06 | 0.2 | 1994 | Beverly Hills—12520 Mulhol | 5.28 | 10.6 | 0.130 | 0.85 |
| 3 | Christchurch—2011 | 0.2 | 2011 | PARS | 5.79 | 8.5 | 0.126 | 0.61 |
| 4 | Sierra Madre | 0.2 | 1991 | Cogswell Dam—Right Abutment | 5.61 | 17.8 | 0.151 | 0.50 |
| 5 | Friuli (aftershock 9)_ Italy | 0.2 | 1976 | San Rocco | 5.5 | 11.9 | 0.127 | 1.41 |
| 6 | Lytle Creek | 0.2 | 1970 | Wrightwood—6074 Park Dr | 5.33 | 10.7 | 0.215 | 1.06 |
| 7 | Christchurch—2011 | 0.5 | 2011 | GODS | 5.79 | 9.1 | 0.175 | 0.63 |
| 8 | Chi-Chi_ Taiwan-05 | 0.5 | 1999 | HWA031 | 6.2 | 39.3 | 0.128 | 1.91 |
| 9 | Chi-Chi_ Taiwan-05 | 0.5 | 1999 | HWA005 | 6.2 | 32.7 | 0.124 | 1.46 |
| 10 | Whittier Narrows-01 | 0.5 | 1987 | Pacoima Kagel Canyon | 5.99 | 31.6 | 0.169 | 1.04 |
| 11 | Chi-Chi_ Taiwan-03 | 0.5 | 1999 | CHY041 | 6.2 | 40.8 | 0.132 | 1.00 |
| 12 | N. Palm Springs | 0.5 | 1986 | Anza—Red Mountain | 6.06 | 38.2 | 0.171 | 1.77 |
| 13 | Chi-Chi_ Taiwan-06 | 1 | 1999 | CHY041 | 6.3 | 45.7 | 0.094 | 0.53 |
| 14 | Northridge-01 | 1 | 1994 | LA—Temple & Hope | 6.69 | 28.8 | 0.113 | 0.62 |
| 15 | Coalinga-01 | 1 | 1983 | Parkfield—Fault Zone 11 | 6.36 | 27.1 | 0.084 | 1.08 |
| 16 | Coalinga-01 | 1 | 1983 | Parkfield—Stone Corral 3E | 6.36 | 32.8 | 0.170 | 1.13 |
| 17 | San Fernando | 1 | 1971 | Lake Hughes #4 | 6.61 | 19.5 | 0.198 | 1.27 |
| 18 | Chi-Chi_ Taiwan-06 | 1 | 1999 | WHA019 | 6.3 | 52.4 | 0.087 | 1.68 |
| 19 | Loma Prieta | 2 | 1989 | SF—Diamond Heights | 6.3 | 71.2 | 0.076 | 0.67 |
| 20 | Chuetsu-oki_ Japan | 2 | 2007 | NGN004 | 6.8 | 78.2 | 0.072 | 1.8 |
| 21 | Chuetsu-oki_ Japan | 2 | 2007 | NGNH28 | 6.8 | 76.7 | 0.051 | 1.80 |
| 22 | Iwate_ Japan | 2 | 2008 | AKT009 | 6.9 | 119.0 | 0.086 | 1.66 |
| 23 | Loma Prieta | 2 | 1989 | Berkeley—Strawberry Canyon | 6.93 | 78.3 | 0.077 | 1.01 |
| 24 | Chuetsu-oki_ Japan | 2 | 2007 | NGNH27 | 6.8 | 91.4 | 0.050 | 1.29 |

## 3. Linear Response Spectrum Analysis

Linear elastic response spectrum analysis is typically taken by contemporary codes of practice as the default method of analysis of the structure. Effects of the higher modes and dynamic torsional actions can be captured by this type of analysis. By applying the principle of modal superposition, the multi-degree of freedom (MDOF) system representing the building is resolved into a number of equivalent single-degree-of-freedom (SDOF) systems with each mode represented by an SDOF system [25,26]. Analysis of each of the equivalent SDOF maximum responses (representing mode 'j' of the MDOF system) is based on solving Equation (1). As Equation (1) has already been solved (for structures of different periods of vibration) while generating acceleration response spectra (Figure 4), once we know the structural period of different modes of vibration, the maximum modal displacement, velocity, and acceleration response of the SDOF system can be easily calculated as '$\omega_{n,j}^2 \, RSA(T_{n,j})$', '$\omega_{n,j} RSA(T_{n,j})$', and '$RSA(T_{n,j})$', respectively. The SDOF modal

responses can then be transformed into MDOF system responses using modal displacement coefficients (representing mode shapes) and combined using modal combination rules.

$$\ddot{u}_j + 2\xi\omega_{n,j}\dot{u}_j + \omega_{n,j}^2 u_j = -\frac{L_j}{M_j}\ddot{u}_g \tag{1}$$

where, $\ddot{u}$, $\dot{u}$, and $u$ are acceleration, velocity, and displacement of the structure measured relative to the ground, respectively; $\xi$ is the damping ratio; $\omega_{n,j}$ is the angular frequency; and $L_j/M_j$ is the participation factor for mode '$j$' and can be represented mathematically as $\{\Phi\}_j^T[M]\{1\}/\{\Phi\}_j^T[M]\{\Phi\}_j$. Here, $\{\Phi\}_j$ is the mode shape vector.

Determining the accurate structural period and displacement coefficient of different modes of vibration is not straightforward, therefore the use of commercial packages such as SPACE GASS [27] is suggested for undertaking the linear response spectrum analysis of the structure. However, the period and the displacement coefficient can be calculated using an approximate method such as in ref. [25], which assumes a specific displacement polynomial shape function. The polynomial shape function ($\psi_j$) for the first, second, and third modes of vibration ($j$ = 1, 2 and 3) of the building supported by shear walls can be approximated by Equations (2a)–(2c), respectively.

$$\psi_1 = 1.5a^2 - 0.5a^3 \tag{2a}$$

$$\psi_2 = 1.7a^2 - 1.95a + 0.03 \tag{2b}$$

$$\psi_3 = -5.4a^5 + 18.3a^4 - 18.7a^3 + 4.4a^2 + a + 0.0011 \tag{2c}$$

where '$a = h/H_e$', '$h$' is the height above ground, and '$H_e$' is the effective height of the building approximately equal to 0.7 × total height, as recommended by Priestley et al. [13].

Similarly, the period of vibration for the first three modes of vibration can be approximated as "$1.72b^{0.5}$", "$0.43b^{0.5}$", and "$0.22b^{0.5}$", respectively, (where '$b = EI/MH^3$', '$E$' is the modulus of rigidity of the concrete, '$I$' is the sum of the gross second moment of area of shear walls, '$M$' is the total mass, and '$H$' is the total height of the building). The first mode approximation is recommended by Chopra [25] and the second and third mode approximations are taken as one-fourth and one-eight, respectively, of the first mode period. Once '$\psi'_j$', '$T'_{nj}$', and '$w_{nj} = 2\pi/T'_{nj}$' are known, the total response spectral displacement can be determined as $\sqrt{\sum_{j=1}^3 \left(\psi_j \omega_{n,j}^2 RSA(T_{n,j})\right)^2}$.

The response spectrum analysis in commercial packages entails the following steps: (1) development of the building model; (2) specifying the cross-sectional and material properties for each structural member; (3) specifying the seismic masses on each floor; (4) specifying the spectral load cases (response spectral curve, the direction of motion, and the critical damping ratio); (5) conducting eigen analysis for determining the dynamic and modal properties of the structure; (6) reading the response spectrum for determining response spectral ordinates corresponding to each mode of vibrations, (7) applying a multiplier ($K_pZ$ for code spectrum and '1' for site-specific response spectrum) for scaling the modal base shear (if required by the code), and (8) applying modal combination (CQC or SRSS).

To ensure that a sufficient number of vibration modes have been incorporated into the analysis and to prevent underprediction of the seismic response, seismic codes typically require the following criteria to be satisfied:

- Total mass participation ratio of at least 0.9 or 90%;
- Consideration of any mode with a mass participation ratio greater than 0.05 or 5%;
- Spectral load cases for the two horizontal orthogonal directions are to be combined using ±100% in the primary and ±30% in the secondary direction;
- Horizontal base shear scaling so that the horizontal base shear obtained from the spectral analysis is not less than a specific percentage of the equivalent static base

shear as stipulated by the code. There is, however, no such requirement stipulated in AS 1170.4-2007 [1].

Response spectrum analysis was implemented on two case study buildings: (I) a 6-storey building, CSB 1 and (II) a 22-storey building, CSB 2. The site-specific mean response spectra and the code-stipulated [1] Site-De spectrum, as presented in Figure 4, were adopted in the analysis. The structural floor plan and member sizes of the two case study buildings are presented in Figures 5 and 6. Structural analysis of the buildings was based on an imposed load of 2 kPa for typical floors, 0.25 kPa for the roof, a superimposed dead load of 1 kPa for a typical floor, 2.5 kPa for the roof, and a façade load of 1 kPa. The seismic weight on each floor of the building was taken as the "total dead load +0.3 imposed load". Information concerning the storey mass and storey height are presented in Table 2.

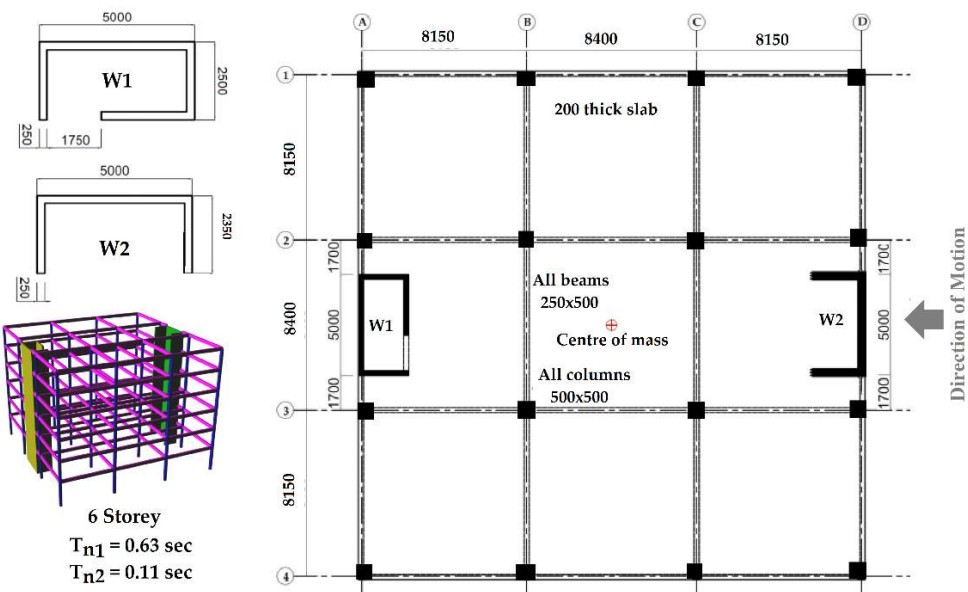

**Figure 5.** Structural layout (showing positions of the frame, and walls W1 and W2), structural detailing, and elevation of case study building CSB 1 (all dimensions are in mm).

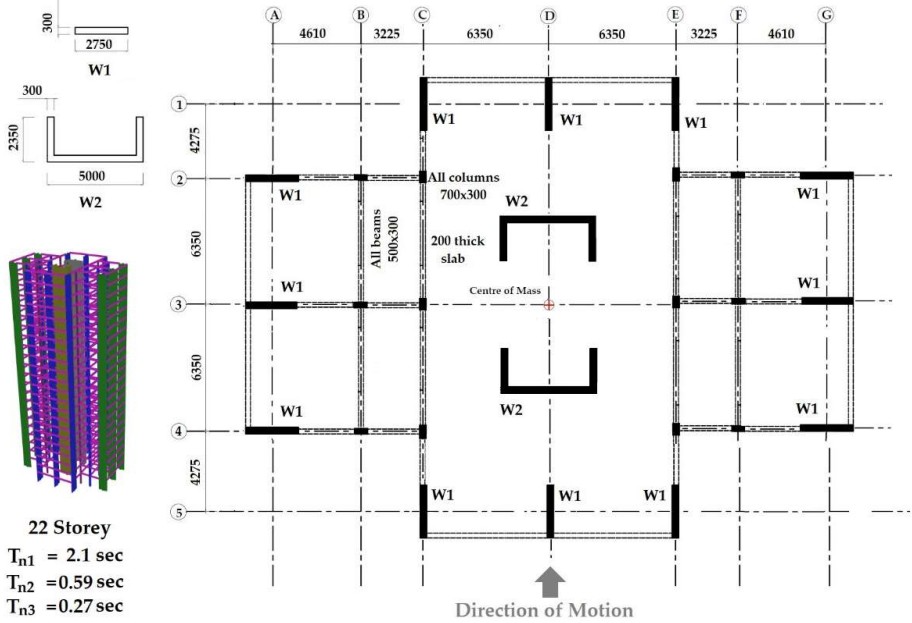

**Figure 6.** Structural layout (showing positions of the frame, and walls W1 and W2), structural detailing, and elevation of case study building CSB 2 (all dimensions are in mm).

**Table 2.** Details of the storey height and storey masses of the case study buildings.

| | Total Height | Ground Floor Height | Other Storey Height | Mass of Ground Floor | Mass of Roof | Mass of Other Storeys |
|---|---|---|---|---|---|---|
| CSB1 | 19.3 m | 3.8 m | 3.1 m | 660 tons | 544 tons | 624 tons |
| CSB2 | 68.9 m | 3.8 m | 3.1 m | 448 tons | 430 tons | 430 tons |

Response spectrum analyses were conducted using the program SPACE GASS Version 12.85 [26]. Results of the modal periods and participating masses as calculated from SPACE GASS are shown in Table 3. Similarly, the storey displacement and design storey shear are presented in Figure 7 for CSB 1 and Figure 8 for CSB 2. The design storey shear was determined by applying a response reduction factor of 2.6, which corresponds to the limited ductility classification.

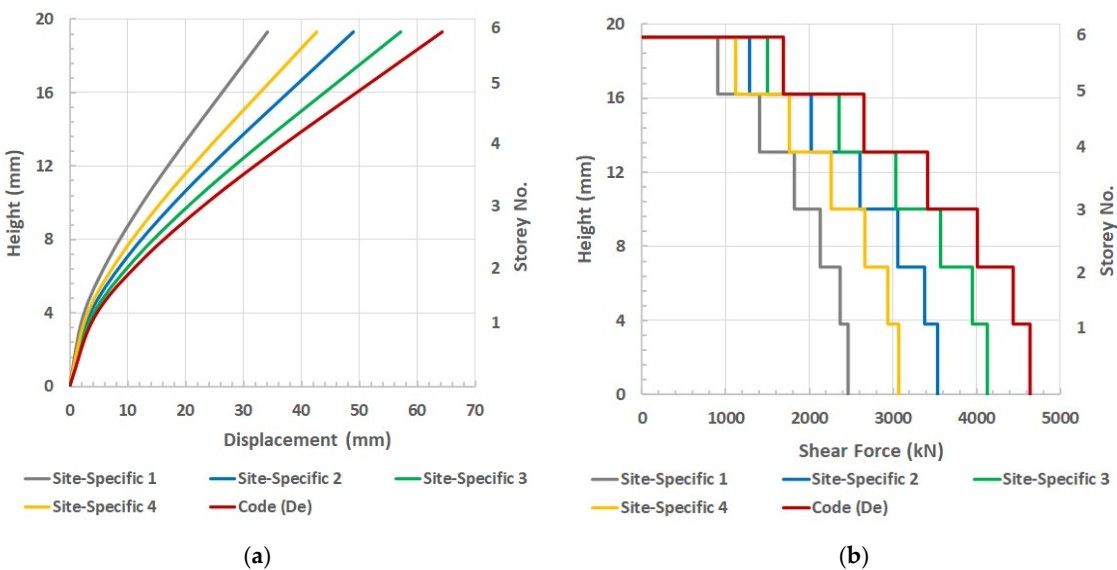

(**a**) (**b**)

**Figure 7.** Analysis results for CSB 1: (**a**) storey displacement and (**b**) storey shear.

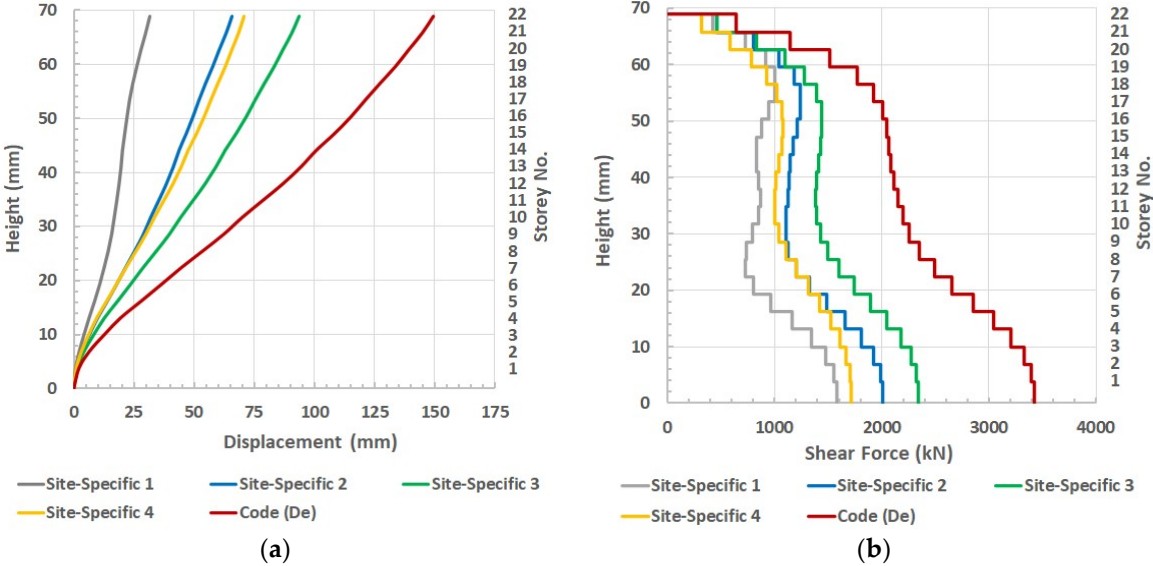

(**a**) (**b**)

**Figure 8.** Storey displacement and storey shear results for CSB 2: (**a**) storey displacement and (**b**) storey shear.

**Table 3.** Modal periods (Tn) and mass participation (MP) for the case study buildings.

| | Modal Periods | | | Mass Participation | | | |
| --- | --- | --- | --- | --- | --- | --- | --- |
| | $T_{n,1}$ | $T_{n,2}$ | $T_{n,3}$ | $MP_1$ | $MP_2$ | $MP_3$ | *Total* |
| CSB 1 | 0.63 s | 0.11 s | - | 69.7% | 20.6% | - | 90.3% |
| CSB 2 | 2.10 s | 0.59 s | 0.27 s | 72.5% | 12.2% | 5.5% | 90.2% |

The critical site-specific spectra have been identified with Site-Specific 3, which corresponds to the reference period T* of 1 s. Site-Specific 3 gave base shear estimates that were about 10% and 30% lower than that calculated from the code spectrum for CSB 1 and CSB 2, respectively. Some 10% and 37% lower estimate of the roof displacement than the code spectrum model was also found for CSB 1 and CSB 2, respectively. The higher difference for CSB 2 than CSB 1 is due to the site-specific spectra being magnified less for the period ranges of interest for CSB 2, which can be seen in Figure 4a,b. It is noted, however, that the site-specific response spectrum can give higher design forces when the fundamental structural period is very low ($T_{n1} < 0.25$ s say) or when the building's natural period is close to the site's natural period.

## 4. Nonlinear Response Spectrum Analysis

A very simple-to-use and computationally efficient macro model-based nonlinear site-specific response spectrum analysis of an RC building is presented in this Section. The macro model employed in this study consists of an equivalent SDOF system with lumped mass at the top and lumped plasticity at the height of "plastic hinge length, $L_p$" measured from the base. The material nonlinearity at this location is defined using a bilinear inelastic force-displacement diagram (the capacity curve) obtained from the simplified pushover analysis procedure. When conditions of nonlinear behaviour occur at any other location, further hinges may form, and the force-displacement diagram would need to be modified to incorporate the effects of additional hinge formation. The response of different elements in support of the building is added to determine the force-resistant capacity of the building as a whole. The displacement capacity of the 'weaker' element can be used for constructing the force vs. displacement diagram, or the acceleration (or force/$M_{eff}$) vs. displacement diagram, representing the whole building. The capacity curve should be superposed with the inelastic site-specific acceleration displacement response spectrum (ADRS) curves representing the seismic demand of the projected earthquake scenario for estimation of the nonlinear response of the equivalent SDOF system. Finally, the response in the floor level (MDOF response) of the building is determined from the SDOF response. The procedure as described is summarised in the schematic diagram Figure 9. Note, only the first mode of vibration has been considered. Thus, the proposed macro model is limited to low to medium-rise RC buildings. Furthermore, the macro-model only accounts for the contribution of the RC structural walls (having vertical reinforcement of 0.5–3.5% and axial load ratio $\leq 0.2$) and neglects the contribution of the gravity frame of the building.

The bilinear force-displacement capacity curve of the individual elements is determined using Equations (3)–(6) as shown below.

At the yield point,

$$\Delta_y = \frac{\phi_y H_e^2}{3} \tag{3}$$

$$F_y = \frac{\phi_y E_c I_{eff}}{H_e} \tag{4}$$

At the ultimate point,

$$\Delta_u = \Delta_y + (\phi_u - \phi_y) L_p \times (H_e - 0.5 L_p + L_{sp}) \tag{5}$$

$$F_u = F_y \tag{6}$$

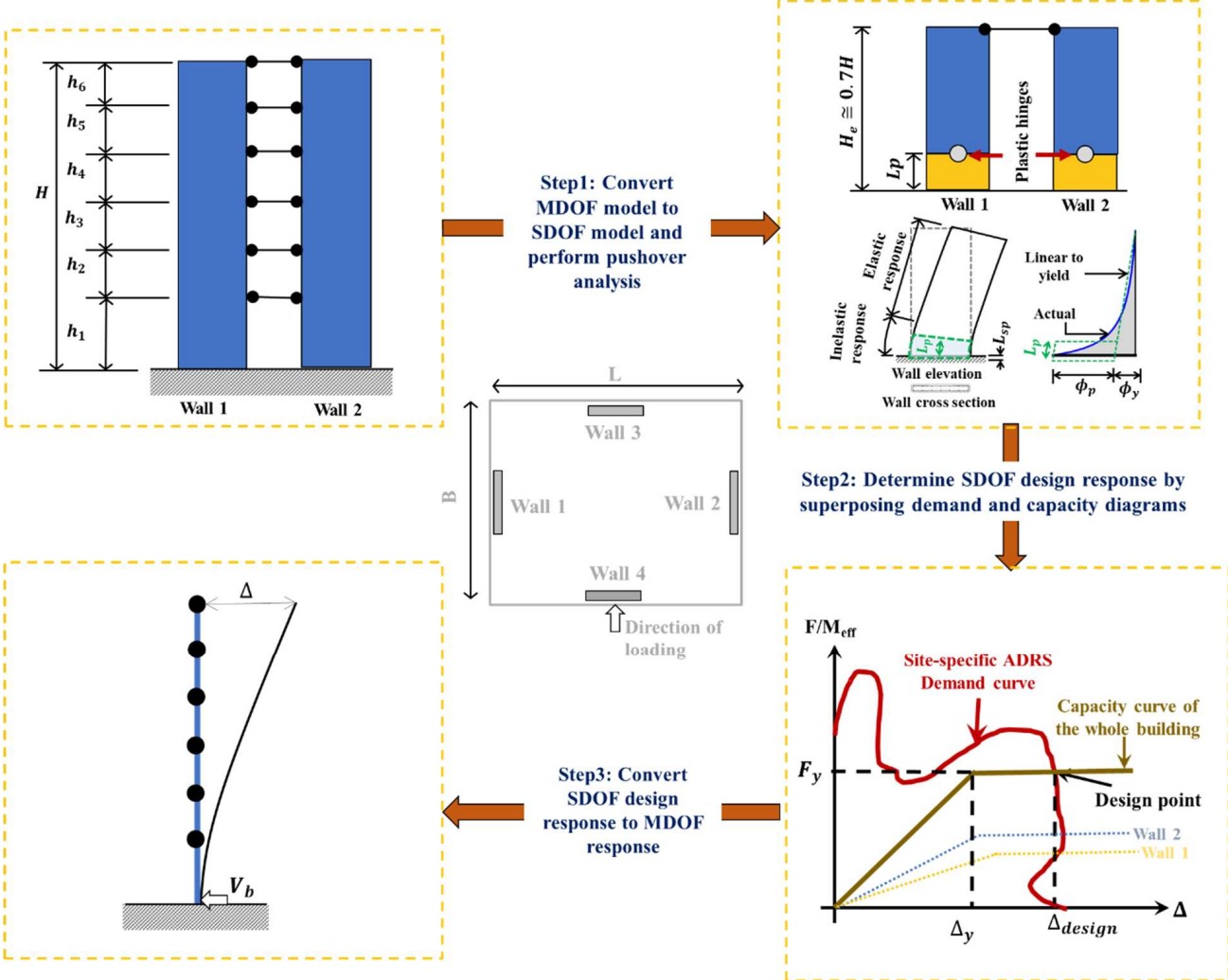

**Figure 9.** Proposed nonlinear analysis procedure.

The yield curvature ($\phi_y$), ultimate curvature ($\phi_u$), effective second moment of section ($I_{eff}$), the overstrength factor ($\Omega$) and ductility ratio ($\mu$) are determined using Equations (7)–(11) given by Menegon [15] as shown below.

$$\phi_y = \left( \frac{b_w L_w^3}{12 I_{gross}} \right)^{0.45} \left( 0.15 p_v - 2 p_v^2 + 0.0031 \right) / L_w \tag{7}$$

$$\phi_u = \left( \frac{b_w L_w^3}{12 I_{gross}} \right)^{0.45} \left[ \left( 19.5 p_v - 545 p_v^2 - 0.066 \right) (0.158 - n) + 0.017 \right] / L_w \tag{8}$$

$$I_{eff} = I_g [ p_v (10 - 30n) + 0.03 n f_{cmi} + 0.1 ] \tag{9}$$

$$\Omega = 9.1 n^2 - 3.6 n + 1.6 \tag{10}$$

$$\mu = \Delta_u / \Delta_y \tag{11}$$

The plastic hinge length ($L_p$) is determined from the plastic hinge model (Equation (12)) as recommended in Priestley et al. [13].

$$L_p = Min \left[ 0.2 \left( f_{su} / f_{sy} - 1 \right), 0.08 \right] \times H_e + 0.1 L_w + L_{sp} \tag{12}$$

where $n$ is the axial load ratio that is equal to axial load / ($f_{cmi} \times$ gross cross sec tional area); $p_v$ is the vertical reinforcement ratio; $L_w$ is the length of the wall; $b_w$ is the thickness of

the web; $I_g$ is the gross moment of area of the wall; $f_{cmi}$ is the mean in-situ strength; $E_c$ is the elastic modulus of concrete; $f_{sy}$ and $f_{su}$ are yield and ultimate stress of reinforcement; $H_e$ is the effective height of the wall (approximately 0.7 of the total height); $L_{sp}$ is yield penetration, which is equal to $0.022 f_{sy} d_b$; and $d_b$ is the diameter of vertical reinforcement.

The displacement at height "$h_i$" from the base is determined from SDOF $\Delta_{design}$ using Equation (13) as shown below.

$$\Delta_{MDOF,i} = \Delta_{y,i} + \Delta_{p,i} = \frac{3}{2}\Delta_y \left( \frac{h_i^2}{H_e^2} - \frac{h_i^3}{3H_e^3} \right) + \left( \Delta_{design,SDOF} - \Delta_y \right) \times \left( \frac{h_i - 0.5L_p + L_{sp}}{H_e - 0.5L_p + L_{sp}} \right) \tag{13}$$

The application of the proposed model is demonstrated with an example case study building. As the proposed method is only applicable to rectangular walls, CSB 1 has been modified by changing the wall configuration as shown in Figure 10.

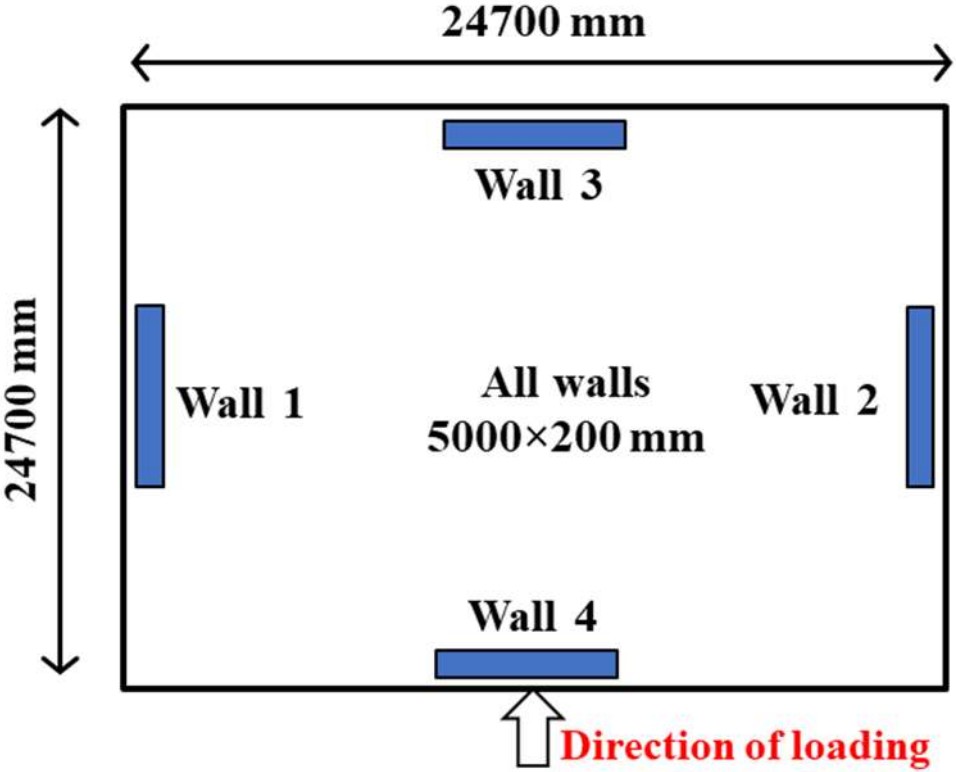

**Figure 10.** CSB 1 was modified with rectangular walls (building height, H = 19.3 m).

The mass and the storey height of the building were kept unchanged. The rectangular walls, which were 5000 × 200 mm in dimensions, were built of 40 MPa concrete ($E_c$ = 32,800 MPa) and 1% of N20 vertical reinforcements ($f_{sy}$ = 550 MPa and $f_{su}$ = 660 MPa). The axial load ratio in each wall was equal to 0.1. By use of Equations (3)–(13), the force and displacement of the structure at first yield, true yield, and ultimate points are calculated as presented below.

$$L_p = Min \left[ 0.2 \left( f_{su} / f_{sy} - 1 \right), 0.08 \right] \times H_e + 0.1 L_w + 0.022 f_{sy} d_b = 1284 \text{ mm}$$

$$\phi_y = 1 \times \left( 0.15 p_v - 2 p_v^2 + 0.0031 \right) / L_w = 8.8 \times 10^{-7} / \text{mm}$$

$$\phi_u = 1 \times \left[ \left( 19.5 p_v - 545 p_v^2 - 0.066 \right) \left( 0.158 - n \right) + 0.017 \right] / L_w = 4.0 \times 10^{-6} / \text{mm}$$

$$E_c I_{eff} = E_c I_g \left[ p_v (10 - 30n) + 0.03n f_{cmi} + 0.1 \right] = 1.95 \times 10^{16} \text{ Nmm}^2$$

$$\Delta_y = \frac{\phi_y H_e^2}{3} = 53 \text{ mm}$$

$$\Delta_u = \Delta_y + (\phi_u - \phi_y) L_p \times (H_e - 0.5 L_p + L_{sp}) = 53 + 52 = 105 \text{ mm}$$

$$F_y = F_u = E_c I_{eff} \phi_y / H_e = 1273 \text{ kN}$$

The total strength capacity of the building was found to be equal to two times the force capacity of the wall, given that the two walls were orientated about the direction of motion having identical dimensions and material properties. The displacement curve represents the capacity of one of the walls. The forces are divided by the effective mass of the structure ($0.7 \times$ total seismic mass = 2591 tons) to determine the corresponding acceleration and to construct the acceleration displacement capacity curve. The superposition of the capacity curve and the inelastic ADRS curve of the AS 1170.4-2007 [1] site $D_e$ and four site-specific response spectra are shown in Figure 11. The inelastic ADRS curves are obtained by modifying the elastic RSA plot (refer to Figure 4), as per Fajfar [10] using Equations (14)–(16), and plotting '$RSD_{inelastic}$' vs. '$RSA_{inelastic}$'.

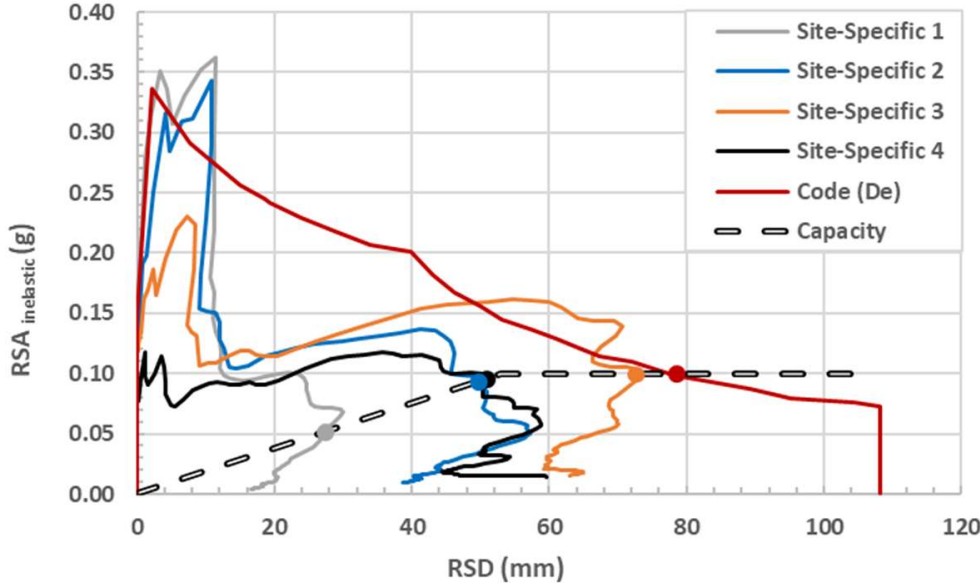

**Figure 11.** Superposition of the capacity curve and the ADRS demand curves (code spectrum and 4 average site-specific spectra).

$$RSA_{inelastic} = \frac{RSA_{elastic}}{R_\mu \Omega} \tag{14}$$

$$RSD_{inelastic} = \frac{\mu}{R_\mu} RSA_{elastic} \left(\frac{T}{2\pi}\right)^2 \tag{15}$$

$$R_\mu = \frac{\Delta_e}{\Delta_y} = \frac{F_e}{F_y} = (\mu - 1)\frac{T}{T_c} + 1 \leq \mu \tag{16}$$

The overstrength factor and ductility ratio are calculated from Equations (10) and (11) as $\Omega = 9.1n^2 - 3.6n + 1.6 = 1.33$ and $\mu = \Delta_u/\Delta_y = 105/53 = 1.98$, respectively, and the corner period '$T_c = 0.53s$' for site class $D_e$ AS 1170.4-2007 [1].

As shown in Figure 11, the maximum SDOF displacement and acceleration values, as read off from the four mean site-specific response spectra, were 74 mm and 0.1 (g), respectively, (refer to 'Site-Specific 3' ADRS spectrum). Similarly, the SDOF displacement and acceleration, as read off from the code spectrum, were 78 mm and 0.1 (g), respectively. The SDOF displacement response is converted to the MDOF displacement response using

Equation (13). The results obtained from the analysis as described are compared with that from elastic analysis as shown in Figure 12.

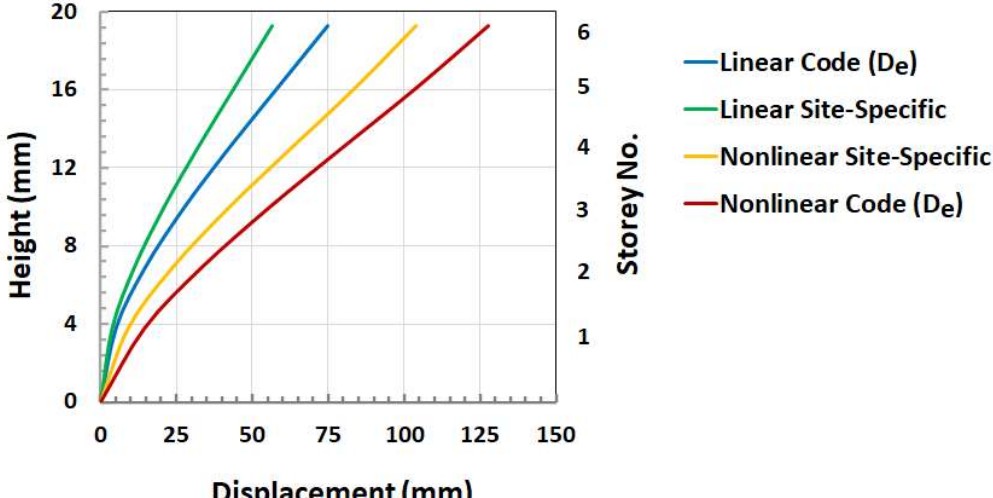

**Figure 12.** Comparison of the floor displacements in modified CSB 1 due to linear and nonlinear site-specific and code response spectrum analysis.

This procedure has neglected the gravity frame of the building (i.e., columns and coupling behaviour of the slabs) for simplicity in illustrating the process. However, depending on the structural configuration and sizing of elements, the gravity frame can affect the dynamic properties of the building and therefore cannot be neglected from the analysis model. Further study is required to provide recommendations in this respect.

## 5. Conclusions

A procedure for deriving site-specific response spectra and the use of the generated spectra to undertake dynamic analysis of a building structure is presented and illustrated with examples. The first step of the procedure is to develop a database of accelerograms on rock outcrops based on a range of earthquake scenarios that are consistent with a predetermined level of seismic hazard in an intraplate environment. Site response analyses are then applied on a representative soil column model of the site, using accelerograms recorded on rock outcrops as input for generating accelerograms on the soil surface of the targeted site. Site response spectra calculated from the generated accelerograms are then averaged to derive the site-specific design response spectra based on different earthquake scenarios. Analysis of the building structure making use of the design spectra is then undertaken. The conventional approach of analysis based on simplifying the structure into a linear elastic system is introduced initially using two case study buildings as examples for illustration. The estimated amount of base shear in the building, as determined from linear elastic dynamic analyses employing the site-specific design response spectra, is shown to be about 30% less than that derived from the code design spectrum. Some 37% less roof displacement is also predicted by adopting the site-specific design response spectra. A more advanced method of assessment involving pushover analysis of a building model with non-linear properties is also illustrated with an example. The strength and displacement demand, as determined from non-linear analysis utilising the site-specific design response spectrum, are also shown to be less conservative than that derived from the response spectrum stipulated by the code for the respective site class.

**Author Contributions:** Conceptualization, P.K.; Methodology, P.K., Y.H., E.L. and S.J.M.; Resources, P.K. and Y.H.; Software, P.K. and Y.H.; Supervision, E.L. and S.J.M.; Validation, P.K.; Visualization, P.K. and Y.H.; Writing—original draft, P.K.; Writing—review and editing, P.K., Y.H., E.L. and S.J.M. All authors have read and agreed to the published version of the manuscript.

**Funding:** This research received no external funding.

**Data Availability Statement:** Not applicable.

**Acknowledgments:** The authors acknowledge the financial support given by the University of Melbourne through its postgraduate research scholarship scheme.

**Conflicts of Interest:** The authors declare no conflict of interest.

## Appendix A. QuakeAdvice: Soil Column Model and Input Information

**Table A1.** Characteristics of soil column obtained from the retrieved borehole record.

| Input Borehole Record Information | | | Output Soil Profile | |
|---|---|---|---|---|
| Thickness (m) | SPT Blow Count | Soil Type | SWV (m/s) | Density (kg/m$^3$) |
| 1.5 | 3 | SC * | 131.7 | 1878 |
| 1.5 | 5 | SC | 152.8 | 1928 |
| 1.5 | 14 | SC | 205.9 | 2025 |
| 1.5 | 14 | SC | 205.9 | 2025 |
| 1.5 | 14 | SC | 205.9 | 2025 |
| 1.5 | 14 | SC | 205.9 | 2025 |
| 1.5 | 14 | SC | 205.9 | 2025 |
| 1.5 | 14 | SC | 205.9 | 2025 |
| 1.5 | 13 | SC | 201.6 | 2025 |
| 1.5 | 11 | SC | 192.0 | 2025 |
| 1.5 | 12 | SC | 196.9 | 2025 |
| 1.5 | 11 | SC | 192.0 | 2025 |
| 1.5 | 12 | SC | 196.9 | 2025 |
| 1.5 | 5 | SC | 152.8 | 1928 |
| 1.5 | 8 | SC | 175.1 | 1928 |
| 1.5 | 10 | SC | 186.8 | 1928 |
| 1.5 | 22 | SC | 234.8 | 2025 |
| 1.5 | 43 | SC | 285.1 | 2145 |
| 1.5 | 72 | SC | 331.1 | 2231 |
| 1.5 | 72 | SC | 331.1 | 2231 |
| Bedrock | - | - | 1000 | 2082 |

* SC stands for sand-clay mixture.

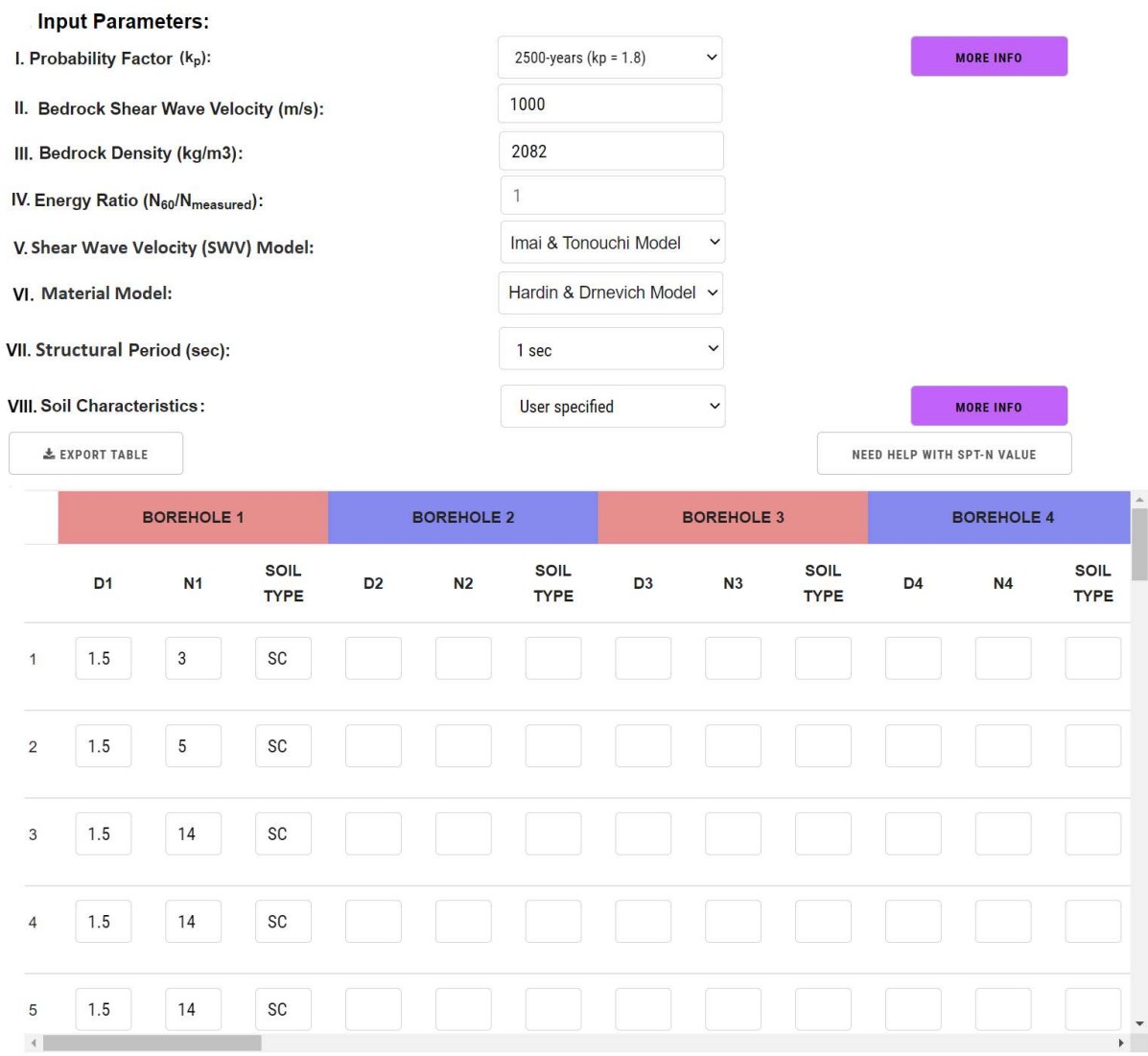

**Figure A1.** Screenshot of the input parameters defined in the "quakeadvice.org" online program [17].

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
