# Peer review of "Dynamic Modal Analyses of Building Structures Employing Site-Specific Response Spectra Versus Code Response Spectrum Models"

_2673-4109, doi:10.3390/civileng4010009_

Round 1

Reviewer 1 Report

The manuscript presents a procedure for deriving site-specific response spectra and the use of the generated site-specific response spectra for the dynamic analyses of building structures. The strength and displacement demands determined from non-linear analysis of case study buildings using the site-specific design response spectrum have been found to be less conservative than that derived from the code-based response spectrum for the respective site class.

The manuscript has been written well. It contains information that is important for the seismic design of structures. Also, the procedure has been explained adequately. The manuscript does not require any major revisions.

Hence, I recommend that the manuscript be accepted for publication.

Author Response

Dear Reviewer,

Thank you for recommending the manuscript for publication in its current form. 

Best regards,

Authors.

Reviewer 2 Report

Please, check my comments in the attached PDF file

Author Response

Dear reviewer,

We really appreciate your efforts and time in providing valuable feedback and comments. Please see our responses to the comments you have highlighted in the pdf file.

Best regards,

Authors

Round 2

Reviewer 2 Report

The paper can be accepted for publication